# Wellbeing across the American Nations: First Settler Effects influence traditional and existential wellness

David R. Samson[1]*, Nathan Oesch[1], Colin Woodard[2]

**1** University of Toronto, Mississauga, Ontario, Canada, **2** Nationhood Lab, Pell Center for International Relations and Public Policy, Salve Regina University, Newport, Rhode Island, United States of America

* david.samson@utoronto.ca

## Abstract

This study investigates the "First Settler Wellness Effect," exploring how cultural geography impacts traditional wellness (physical health, social relationships, and financial stability) and existential wellness (purpose, meaning, and community identity) across the United States. Using data from the Gallup-Healthways Well-Being Index, which includes responses from over 325,000 individuals across 110 Metropolitan Statistical Areas (MSAs) from 2009 to 2016, we analyze wellness outcomes through the lens of the American Nations Model. This model categorizes the United States into distinct cultural regions shaped by early settlement patterns, emphasizing the enduring influence of regional norms and ideologies. Our findings, in support of the American Nations Model, reveal significant regional variation in wellness outcomes. Northeastern and Midwestern regions, characterized by communal norms, high educational attainment, and institutional trust, exhibit elevated traditional wellness scores (e.g., $\beta = 0.371$, $p = 0.002$). These regions reflect a stability rooted in health infrastructure and economic security. In contrast, Southern regions, shaped by honor-based values emphasizing personal autonomy, loyalty, and social reputation, show significantly higher existential wellness (e.g., $\beta = 0.590$, $p = 0.011$). This divergence highlights a tradeoff between material stability and existential fulfillment shaped by cultural norms. Interestingly, Southern regions demonstrate elevated existential wellness for Black and Hispanic residents compared to other regions, suggesting localized cultural or community support may offset systemic disparities. Conversely, Northeastern and Midwestern regions report higher traditional wellness yet fail to foster similar levels of existential fulfillment, underscoring the limitations of material prosperity alone. These findings emphasize the interplay between cultural history, regional identity, and human flourishing, offering insights for targeted public health and policy interventions.

**Data availability statement:** The full dataset, along with all meta-data and detailed descriptions of each variable, is available in the Open Science Framework (OSF) data repository: https://osf.io/jxsz8/?view_only=-968233c39a594d14bed3f201b24ec0f8.

**Funding:** This research was supported through Discovery Grants from the Natural Sciences and Engineering Research Council of Canada (RGPIN-2020-05942 to D.R.S.). Additional support was provided by the Department of Anthropology, University of Toronto Mississauga and The Pell Center's Nationhood Lab.

**Competing interests:** The authors have declared that no competing interests exist.

## Introduction

The question of what drives life satisfaction, happiness, and purpose – hereafter collectively referred to as "wellbeing" – has long intrigued scholars and thinkers across various disciplines, including philosophy, psychology, religion, and economics [1]. The question of whether money and material wealth can indeed "buy happiness" has been debated for millennia, as far back in history as the ancient Greek philosopher, Aristotle [2]. More recently, American psychologist Abraham Maslow identified basic material needs as foundational to wellbeing in his hierarchy of needs [3]. Yet, though widely believed that good health is essential for happiness, emerging psychological research indicates that people generally adapt and can find ways to recover their happiness even after serious health issues, such as following the recovery of a serious illness or loss of a limb [4]. Accordingly, a growing body of recent research has been steadily indicating that while health, money, and security are inarguably important, such factors are unlikely to be the only drivers of wellbeing. For example, recent cross-cultural work has further demonstrated that cultural forces, such as group identity, community, social relationships, lack of corruption, and freedom can increase happiness, even when taking economic development and national wealth into account [5].

### Traditional and existential measures of wellbeing

Interestingly, recent studies on life satisfaction and happiness suggest that wellbeing plays a more significant role in health than previously understood [6–8]. Building from an initially incomplete definition and conceptualization of wellbeing, Forgeard and colleagues [9] have proposed broadening previous indices to include dimensions such as pleasure (subjective well-being), engagement, good relationships, meaning and purpose, and accomplishment, collectively known as PERMA. As such, combining both subjective and objective indicators, in this way, to construct this multidimensional construct of wellbeing creates a much more comprehensive and objective measure. Encouragingly, both national and international survey instruments have recently been incorporating these measures, reflecting a shift from a narrow focus on happiness to a multifaceted approach to well-being, providing a much richer index capable of capturing various dimensions of human flourishing [10].

On the one hand, traditional wellbeing encompasses quantifiable aspects of health and quality of life, including physical health, mental health, economic stability, and social well-being. Namely, physical health pertains to fitness and absence of disease, while mental health involves emotional stability and overall happiness [11,12]. On the other hand, economic stability denotes financial security and freedom from severe stress. For instance, economists have historically argued that positive wellbeing is a simple consequence of greater income. And indeed, this is undeniably true; especially on a global scale, higher incomes lead to reductions in extreme poverty and disease, improvements in food diversity and security, increases in school attendance, and increased health care expenditures [13]. Moreover, greater household income is typically associated with better health measures and outcomes such as normal

height, waist-to-hip ratios, respiratory function, and malaise, thereby limiting the occurrence of acute or chronic illnesses [14]. Thus, poverty alleviation is associated with happier populations, and recent work has borne this out, in showing that high or volatile inflation is generally detrimental for a population's wellbeing, especially among those with a right-wing political orientation within the United States [15,16].

Studies by economists Daniel Kahneman, Angus Deaton, and others have found that, once income has reached a basic subsistence level, namely, that of a pre-tax annual family income of around $75,000, such effects rapidly plateau [17]. From this research, one could conclude that although to a point, money does appear to buy happiness, this non-trivial effect on wellbeing appears to quickly diminish, once a basic income threshold has been reached [18]. Yet, more recent research complicates the picture. A large-scale experience-sampling study found that well-being continues to rise with income across the spectrum, including above $75,000, but in a logarithmic rather than linear fashion—meaning that proportional increases in income, not absolute gains, are consistently associated with higher well-being [19]. This suggests that while marginal returns may diminish, the positive relationship between income and subjective well-being remains robust and continuous at all income levels.

In contrast, existential wellbeing addresses deeper, less tangible aspects of life, such as meaning, purpose, and fulfillment [20]. It includes having meaningful goals, personal growth, community identity and strong community networks, living authentically according to one's values, and existential security in understanding life and existence [21]. For instance, distinct from economic stability, social well-being refers to both the quality and quantity of social relationships and social networks [22]. In particular, social relationships have been shown to have a profound influence on wellbeing, health and human disease [23–25]. For example, among people with fewer sources of social and emotional support, such individuals typically have higher rates of mortality and morbidity, especially from angina and similar cardiovascular issues, likely related to higher blood pressure, cortisol levels, and overall difficulties coping with stress [25]. Indeed, research has shown the presence of a friend in a stressful situation can often be linked to lower blood pressure [26], while embraces from a loved one prior to a stressful situation commonly stimulates oxytocin production, acting as an endogenous analgesic. Beyond this, the influence of social networks on health and human disease can be even more profound, influencing the spread of a wide variety of behaviors and psychological states, including negative conditions such as obesity [27], smoking [28], and depression [29], as well as positive states, such as informing the prevention of social isolation and loneliness [30] and facilitating happiness [31].

From a less personal, more political perspective, additional studies have shown that federalism and democracy generally facilitate strong feelings of wellbeing in individuals [32]. Specifically, the more direct the political participation opportunities which have become available to citizens, the greater their wellbeing. Firstly, a more active role in politics by the citizenry allows for better monitoring of politicians and policy makers by average citizens, typically leading to greater approval of government outcomes. Secondly, the capacity for citizens to have control over and get involved in their own political decisions and outcomes – for example, by way of local public referendums – further raises wellbeing.

Fortunately, in recent years, these and various other similar wellbeing indicators have been increasingly measured through surveys and clinical assessments, and statistically evaluated across a wide array of populations. It stands to reason, then, that different geocultural locations throughout North America and the United States are characterized by different social norms and values that are associated with different levels of both traditional and existential wellbeing.

## First Settler Effects

Woodard [33] expanded upon the "Doctrine of First Effective Settlement" [34], which posits that the first successful settlers of a newly-colonized region have an outsized influence on the future trajectory of that society, including contemporary social, cultural, and ideological characteristics. Regional cultures can thus be discerned and mapped by tracking successful colonizer-settler patterns, an approach that has informed the work of historians [35,36], ethnographers [37], linguists [38], political scientists [39] and geographers [40–42]. Anthropologists have described how culture persists over long time

periods, via intergenerational transmission, conducted through passive observation, imitation, shared participation, and formal teaching within families, schools, places and forms of worship, community organizations, ritualistic ceremonies and other social contexts [43–45]. Psychologists have shown how individual members of nations and other large social groups share collective – and often highly selective or invented – stories of the past; the individuals are the agents doing the remembering of these stories, but the content is shaped by their sociocultural environment [46–48].

## The American Nations Model

Woodard's American Nations framework delineates distinct cultures across North America north of the 25th parallel, including thirteen regional cultures or "nations" and the "anomalous" Federal Entity (the District of Columbia) [49]. Among these nations, some parallel regions identified by Fischer [35], such as Yankeedom, were originally settled by English Puritans who believed they had been tasked by God to create a more perfect society and, thus, strongly emphasized communal success over individual liberty, and Greater Appalachia, populated largely by Scots-Irish immigrants known for their fierce defense of individual autonomy and personal liberty and resistance to external control by government or most anyone else [50]. Similarly, Woodard's El Norte aligns with Garreau's MexAmerica [51], and his Left Coast mirrors Garreau's Ecotopia, a concept initially inspired by literary works [52]. Empirical studies have demonstrated that Woodard's nations exhibit distinct differences in various domains, including entrepreneurship [53], economic development [54], mortality rates [55], obesity, diabetes, and physical inactivity [56], social vulnerability [57], and voting patterns [58]. Like Fischer's model, Woodard's taxonomy is deeply rooted in historical settlement patterns, yet it also offers a comprehensive overview of contemporary societal structures. Fig 1 presents a map of Woodard's model, with Table 1 providing brief descriptions of each nation. The methods Woodard used to trace the boundaries of these cultures have been described elsewhere [33,49], and the history and cultural characteristics of each region have been discussed in detail in four book-length treatments focused on their history [33], communitarian-individualistic orientation [59], relationship to competing narratives of United States nationhood [60], and specific contemporary policy issues [61].

## Regional geocultural coalitions

When considering clusters of regional cultural characteristics among the nations, the Northeastern and Midwestern nations include Yankeedom, New Netherland, and the Midlands. These regions have been described as *dignity cultures*. Dignity cultures, which emerged from Enlightenment liberalism, prioritize the inherent worth of individuals regardless of their social standing. In dignity cultures, personal dignity is seen as an inalienable right, and individuals are encouraged to resolve disputes through legal and institutional channels rather than personal retribution. This fosters a more egalitarian social structure where respect is granted based on intrinsic human rights rather than external validation [62]. Conversely, the more southern regions (Greater Appalachia, Deep South, and Tidewater) are described as *Honor cultures.* Honor cultures, prevalent in societies where individuals must defend their reputation through personal bravery and adherence to communal norms, emphasize external recognition and social standing. In these cultures, respect and status are often maintained through displays of strength, loyalty, and the willingness to retaliate against insults or slights. This leads to a collective enforcement of social norms, where the community plays a significant role in regulating behavior and resolving conflicts.

Anthropologically, honor cultures have often arisen among herding societies, where the mobility and wealth associated with livestock necessitate a strong emphasis on reputation and communal enforcement of norms. The Sami of Northern Europe, the Tuva of Central Asia, and the Scottish Lowlanders (whom are the progenitors of a vast majority of the settlers of Greater Appalachia) all exemplify honor-based cultures shaped by their herding lifestyles. Among the Sami, herding reindeer in the harsh Arctic environment, and maintaining honor through bravery and communal solidarity is crucial for survival and protection against external threats [63]. Similarly, the Tuva, nomadic herders of Siberia, emphasize honor in their social structure, valuing personal bravery, loyalty, and the defense of their herds

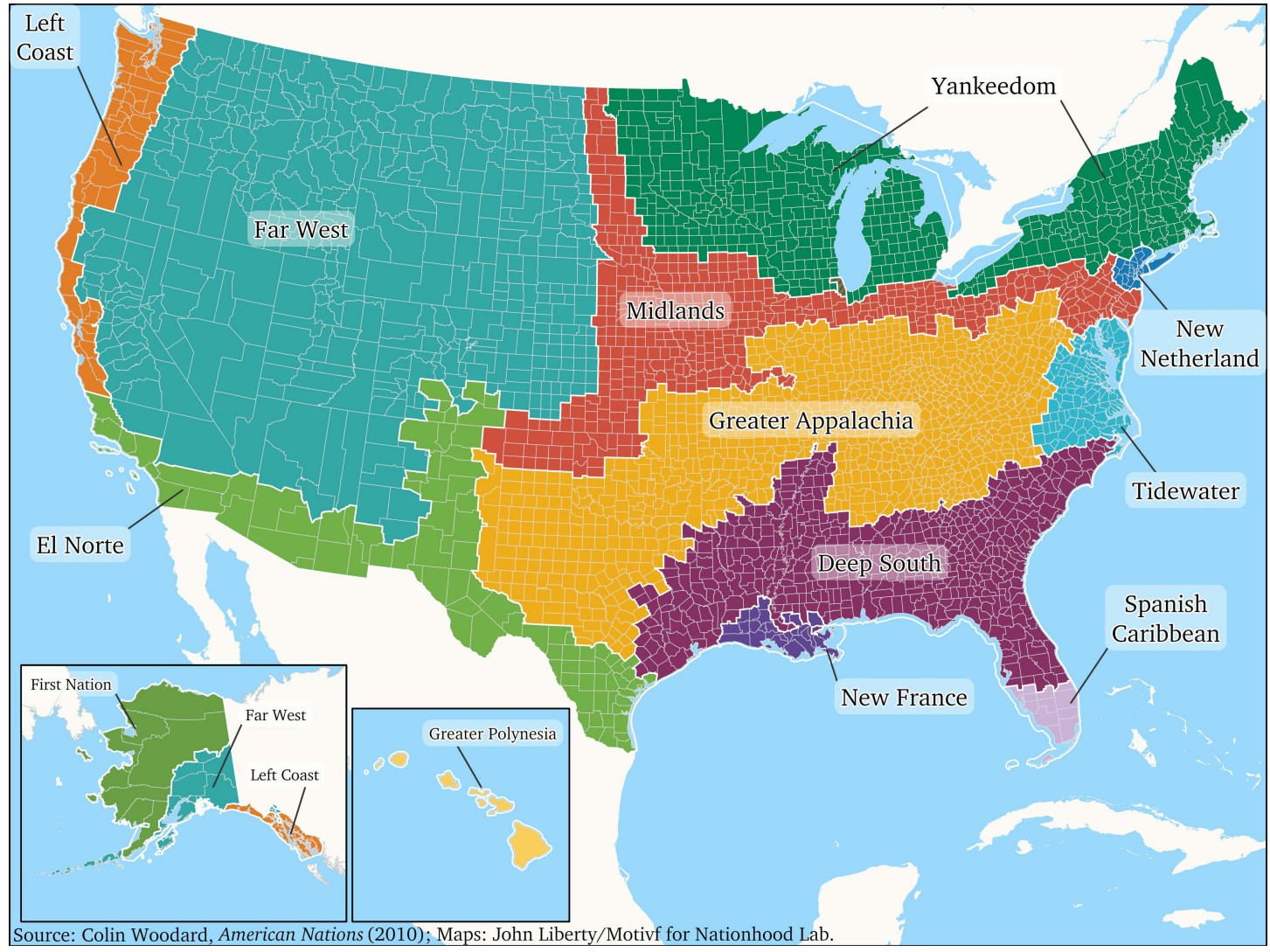

**Fig 1. This map illustrates the regional division of North America into distinct cultural nations as proposed by Colin Woodard in his 2011 book "American Nations: A History of the Eleven Rival Regional Cultures of North America".** The model divides the continent into several "nations," each with its own unique cultural identity shaped by the settlement patterns, historical events, and prevailing ideologies of the first settlers in those regions.

and territory [64]. The Scottish Lowlanders with their herding of cattle and sheep in contested and sometimes lawless border regions, developed a culture of honor that prized martial valor, kin loyalty, and the protection of clan reputation. These herding societies, including the Maasai of East Africa, who also prioritize honor through cattle raiding and warriorhood, showcase how the demands of pastoral life fostered cultures where honor is essential for social cohesion and survival [65].

Crucially, while herding itself has largely declined in many of these regions, the underlying cultural patterns persist due to well-established processes of cultural transmission. Through vertical (parent-to-child), horizontal (peer-to-peer), and oblique (institutional) mechanisms, honor-based values continue to be passed down and reinforced. Local institutions—such as churches, schools, and community organizations—serve as cultural vectors that embed and perpetuate these

**Table 1. Descriptions of Woodard's American Nations with region and average MSA sample size.**

| Nation | Description | Region | MSA Average Sample Size |
|---|---|---|---|
| **Northeastern and Midwestern nations** | | | |
| Yankeedom (YK) | Settled by educated Puritans, characterized by an appreciation for education and strict community oriented, social norms of a covenanted people. | New England and Great Lakes | 964 |
| New Netherland (NN) | Diverse, commercial, and tolerant, reflecting its Dutch origins. | Greater NYC | 4821 |
| The Midlands (ML) | Pluralistic and pacifist, originally established by English Quakers. | Industrial East and Midwest | 860 |
| **Southern nations** | | | |
| Tidewater (TW) | Established by younger sons of southern English gentry, with a conservative, honor based hierarchical structure. | North Carolina to Delaware | 1205 |
| Greater Appalachia (GA) | Populated by Scots-Irish honor culture immigrants, marked by independence and skepticism of government. | North Texas to West Virginia | 906 |
| Deep South (DS) | Founded by English slave lords from Barbados, exhibiting a rigid social structure, and a hostility to the notion of human equality. | East Texas to South Carolina | 801 |
| New France (NF) | Originally settled by French colonists, maintaining a unique French-speaking culture in North America. Historically open to communitarian social democracy at the local level. | Southern Louisiana (and Quebec in Canada) | 652 |
| **Western nations** | | | |
| El Norte (EN) | Northernmost part of New Spain's imperial project in the Americas, so remote that it developed characteristics distinct from central Mexico. Strong adherence to traditional culture and social norms. | Southern California to South Texas | 1557 |
| The Left Coast (LC) | Settled by a blend of New Englanders and prospectors from Greater Appalachia known for its mix of utopian and entrepreneurial ideals. | Pacific Northwest, west of the Cascades | 1143 |
| The Far West (FW) | Characterized by a strong individualism, with initial settlement driven by mining and railroads. | Great Basin to Western Plains | 591 |
| **Small nations** | | | |
| Spanish Caribbean (SC) | Northernmost extension of Spain's maritime imperial cultural space, with primary hub in Havana. | South Florida (and Cuba and parts of Greater Antilles) | 1304 |
| Greater Polynesia (GP) | Hawaii and other Pacific islands including regions predominantly settled by Polynesian long-distance navigators; it is characterized by strong cultural traditions, seafaring heritage, and community-oriented societies. | U.S. region including Hawaii rooted in Pacific culture, founded by small-scale oceanic navigators | 325 |
| Federal Entity (FE) | An administrative region created by and for the federal government, with structures and norms evolving independently from neighboring regions. | District of Columbia proper | N/A |

norms, even in the absence of the original subsistence ecology. An agent-based model supports this, showing that honor cultures thrive where formal authority is weak, reinforcing norms of self-reliance and group protection [66]. These insights help explain why Western "dignity" strategies often falter in honor-based regions: the psychological and institutional legacy of early settlement patterns continues to shape regional wellness outcomes today.

## Hypotheses and predictions

In this study, we investigate three primary hypotheses related to cultural geography and its impact on wellbeing. The **Settler Wellness Effect hypothesis** posits that cultural geography and first settler effects significantly influence an individual's form of wellbeing, encompassing not only traditional metrics (physical, social, and financial wellbeing) but also existential wellbeing, which includes a sense of purpose, meaning, and community identification. We predict that the American Nations regions, based on their cultural norms and differences, will exhibit distinct wellbeing outcomes. The

**Honor-Based Wellness hypothesis** asserts that American Nations with honor-based cultural norms will demonstrate greater existential wellbeing but lower traditional wellbeing and life satisfaction.

The **Minority Identity Disparity Hypothesis** suggests that non-majority racial out-groups will experience lower levels of wellness and life satisfaction compared to majority in-groups across the various cultural regions of the American Nations. In the Northeastern and Midwestern nations, we anticipate disparities in wellbeing between White and non-White populations, with particularly pronounced gaps for Black populations. In the Southern nations, we predict significant disparities, with non-White populations facing markedly lower wellness due to the region's historical racial hierarchies. In the Western nations, the dynamics are more complex: in El Norte, Hispanic populations are expected to experience higher wellness, while non-Hispanic groups may face lower levels; in The Left Coast and The Far West, White populations are likely to maintain higher wellness, with potential disparities for non-White groups. This framework provides a nuanced understanding of how regional, cultural and historical contexts shape contemporary disparities in wellbeing.

## Methods

### Participants and data collection

The data for this study were derived from the Gallup-Healthways Well-Being Index, which has been tracking the well-being of U.S. adults since 2008 through a comprehensive survey methodology. This index provides a nearly real-time view of Americans' well-being, offering insights into five interrelated elements: sense of purpose, social relationships, financial security, community relationships, and physical health [66]. This research was conducted in accordance with the ethical standards of the University of Toronto's Research Ethics Board, following guidelines for human participant research and data confidentiality.

Gallup-Healthways interviews U.S. adults aged 18 and older living in all 50 states and the District of Columbia using a dual-frame design that includes both landline and cellular telephone numbers [67]. The survey employs random-digit-dial methods to sample landline and cellphone numbers. Within households, Gallup selects respondents at random based on the member who had the most recent birthday. Each national sample includes a minimum quota of 50% cellphone respondents and 50% landline respondents, with additional minimum quotas by time zone within each region. Interviews are conducted in Spanish for respondents who are primarily Spanish-speaking. Gallup conducts interviews with at least 500 U.S. adults daily, accumulating over 175,000 interviews annually. Since its inception in 2008, the survey has conducted more than 2 million interviews. These extensive data collection efforts ensure that the analysis captures a wide range of factors affecting well-being, allowing for an in-depth examination of how cultural geography and settlement patterns influence individual and community well-being across the United States.

In addition to wellness data, the study incorporates various demographic and socioeconomic variables from public datasets, such as racial diversity, educational attainment, political leanings, religious diversity, and income levels, to provide a comprehensive understanding of the factors influencing well-being across different American Nations. The dataset for this study includes yearly measures of wellness for Metropolitan Statistical Areas (MSAs) from 2009 to 2016, providing a robust longitudinal dataset for analysis.

### Ethics statement

This research was conducted in accordance with the ethical standards of the University of Toronto's Research Ethics Board. The study adhered to all institutional and national guidelines for research involving human participants, including the principles outlined in the Tri-Council Policy Statement on Ethical Conduct for Research Involving Humans. All data were anonymized and aggregated prior to analysis, ensuring participant confidentiality and data protection. No identifiable information was used, and ethical approval was obtained where required for secondary data analysis.

## American Nations Model

MSA's were coded into their respective American Nations Model geography [33]. Table 1 describes geo-cultural characteristics, as well as the associated average respondent number for each nation. In order to assess the characteristics of communities, we initially aggregated individuals into MSAs. However, the number of respondents in most MSAs varied: Of the 110 MSAs represented in the data, several had smaller sample sizes. These small MSAs, however, included only a minor portion of the total sample. This led to a more robust dataset which included a total of 325,777 individuals. These averages were derived from the responses collected over the years from 2013 to 2016, providing a comprehensive overview of the sample distribution across different American Nations.

## Data, factors, & variable transformation

The response variables for this study were derived from components of wellbeing as measured by respondents in the U.S. Dailies Gallup survey, reported as aggregate averages across MSAs. These variables encompass both traditional and existential dimensions of wellbeing. As defined by Gallup-Sharecare, traditional wellness variables, encompassing physical, social, and financial wellbeing, are represented by indices that capture the more measurable aspects of an individual's health and quality of life. Although social well-being could reasonably be associated with existential dimensions, we classified it within the Traditional Wellness Index, due to extensive evidence linking the quality of social relationships—particularly loneliness and social isolation—to concrete physical health outcomes, such as cardiovascular disease, immune function, and mortality risk, aligning it more closely with other material and physiological dimensions of wellness.

1. *Physical wellbeing* includes metrics related to having good health and sufficient energy to perform daily tasks.

2. *Social wellbeing* reflects the presence of supportive relationships and love in one's life.

3. *Financial wellbeing* involves managing one's economic life to minimize stress and enhance security.

   In contrast, existential wellness addresses deeper, philosophical aspects of human life.

4. *Community wellbeing* measures the extent to which individuals like where they live, feel safe, and take pride in their community, capturing a sense of belonging and security.

5. *Purpose wellbeing* relates to having meaningful goals and feeling motivated to achieve them, reflecting a sense of direction and personal fulfillment.

The *Traditional Wellbeing Index* is calculated as the Z-score of physical, social, and financial wellbeing, integrating these elements to provide a comprehensive measure of traditional health and quality of life. The *Existential Wellness Index* is calculated as the Z-score of community and purpose wellbeing, capturing the more intangible aspects of meaning, purpose, and community identification. Together, these indices provide a holistic view of wellbeing, encompassing both the quantifiable and philosophical dimensions of human experience.

The *Regions* factor categorizes the *American Nations* into broader geographical regions as outlined in Table 1. This classification helps to contextualize the diverse cultural and social dynamics within each American Nation. Specifically, the Regions factor groups these nations into the following categories: Northeastern and Midwestern Nations – which for the purposes of testing the second hypothesis – are labeled as *Dignity Culture,* Southern Nations are labeled as *Honor Culture,* and remaining regions including the Western and Small Nations. This regional classification reflects the historical and cultural contexts that shape the wellbeing outcomes within these distinct areas.

The *Racial Diversity (Entropy Index)* was computed from demographic data detailing the racial composition within multiple MSAs in the United States. This process involved converting racial percentages into proportions. The racial composition was calculated proportionally, reflecting the distribution of Non-Hispanic Whites, Blacks, Hispanics, Asians, and others

within each MSA. For the Entropy Index, we then utilized Shannon's entropy formula [51]. The entropy index is a measure of uncertainty or randomness in the racial distribution within an MSA, with higher values indicating greater diversity (more even distribution across racial groups) and lower values suggesting less diversity (dominance by one or a few groups). This index provides a nuanced understanding of racial diversity, facilitating comparative analyses across different urban settings and offering insights into the dynamics of racial composition.

The *Education Index* was derived by first converting survey response percentages into numeric values. Educational categories were assigned specific values reflective of their perceived educational duration: "High school or less" was given a value of 3, "Some college" a value of 6, "College graduate" a value of 9, and "Post graduate" a value of 14, as representative of the years put towards educational attainment. These values were then aggregated, providing a composite score that quantified educational attainment for each observation in the dataset.

The *Political Party Continuum* serves as a quantitative measure to assess the relative balance between conservative and liberal political orientations within MSAs or regional contexts. This index typically ranges from negative to positive, with negative values indicating a conservative-leaning sentiment and positive values reflecting a liberal-leaning sentiment (where zero equals perfectly moderate political orientation). The index provides insight into the predominant political affiliation within the region and the intensity of these leanings, with higher positive values suggesting a more pronounced liberal-leaning population and lower values indicating a stronger conservative orientation. For example, the presence of a significant number of independents would moderate the index, resulting in a more balanced or centrist political landscape.

A measure of *Religious Diversity* was calculated by synthesizing data which detailed religious affiliations across various geographic regions. This process involved converting percentage representations of religious affiliations into numeric values and applying a custom function to calculate the entropy index. The index measures the diversity and distribution of religious affiliations, with higher values indicating greater diversity (more even distribution among different religious categories) and lower values suggesting lower diversity (dominance by one or a few categories).

The *Income Index* was developed to provide a comprehensive ordinal measure of income distribution. Each income bracket was assigned an ordinal value: < \$3,000 per month = 1, \$3,000 − < \$7,500 per month = 2, and \$7,500+ per month = 3. Percentages for each income bracket were converted to proportions and used to compute the ordinal index through a weighted summation approach. This index captures the distribution of income within the dataset and serves as a nuanced metric for examining income effects in quantitative analyses. *Population Data* reflects the average number of respondents for wellness surveys per MSA, providing a proxy for MSA population size. *Religious Importance* was measured by the proportion of respondents who indicated that religion is important to them. This variable was derived from responses categorized as "yes," "no," or "don't know (DK)."

## Modeling

To assess the predictors of the five response variables—Physical, Social, Financial, Community, and Purpose wellbeing—within the American Nations Model, where the American Nations (or regions) served as the factor of analysis, we employed a linear model in R 4.4.0 [68]. To normalize response variables, all continuous variables were scaled. Statistical inferences were made using a combination of standardized coefficients, confidence intervals, and p-values.

For Hypothesis 1, our primary factor of analysis was the fixed effects of *American nations*. The American nation of *Yankeedom* was used as the reference category, serving as the point of comparison for other groups in the statistical analysis. Consequently, estimates for each population represent predicted differences in wellness response counts compared to this reference sample. To explicitly assess whether cultural regions differentially influence traditional versus existential well-being, we conducted a within-subjects linear mixed-effects analysis. In this framework, the two wellness indices (Traditional and Existential) were concatenated into a single outcome variable, accompanied by a binary indicator for wellness type. The model included fixed effects for wellness type, American Nation, and their interaction—thereby testing whether the effect of each nation on well-being depends on whether one considers material (traditional) or meaning-oriented

(existential) dimensions—while controlling for education and income. A random intercept for each metropolitan area accounted for non-independence of observations. For hypothesis 2, our primary factor of analysis was the fixed effects of *American nations* grouped into Regions (see Table 1). The Dignity culture of the *Northeastern and Midwestern nations* is the reference group. For hypothesis 3, we performed a linear mixed-effects model where MSA was a repeated measure. Our primary factor of analysis was the fixed effects of *American nations* and *Racial demographics*, while controlling for *Income.*

Model fitting was performed to assess multicollinearity. The correlation matrix highlighted significant associations among key variables in the wellbeing assessment. *Political orientation* and *Religious diversity* showed a strong positive correlation (r = 0.71), while *Racial diversity* had a moderate positive correlation with *Political orientation* (r = 0.35). *Education* and *MSA population size* exhibited a weaker correlation (r = 0.28). Variance Inflation Factors (VIF) were computed to assess multicollinearity among predictors in each Wellness Index regression model. The *American Nations* displayed a high GVIF of 8.996, indicating substantial multicollinearity among its categories, suggesting overlapping effects on the wellness index. *Religious diversity* and *Religious importance* also showed high GVIF values around 7. To address these concerns, religious importance was removed from the final model, and the *Racial diversity* was excluded to focus on race (categories) in subsequent analyses. With these adjustments, no remaining predictors exhibited high GVIF values, enhancing the model's stability and interpretability. This underscores the American Nations Model's robustness in predicting wellness.

The models to test hypothesis 1 were specified as follows:

**Model 0:** Wellness response ~ American Nations Wellness

Model 1: Wellness response ~ Wellness type * American Nation + EducationIndex + IncomeIndex + (1 | MSA)

**Model 2:** Wellness response ~ Racial diversity + Political orientation + Education + Religious Diversity + Income

The models to test hypothesis 2 were specified as follows:

**Model 3:** Wellness response ~ Region Wellness

**Model 4:** Wellness response ~ Racial diversity + Political orientation + Education + Religious Diversity + Income

The models to test hypothesis 3 were specified as follows:

**Model 5:** Wellness response ~ Racial demographic + Income + Region + (1 | MSA)

The full dataset, along with all meta-data and detailed descriptions of each variable, is available in the Open Science Framework (OSF) data repository: https://osf.io/jxsz8/?viewonly=968233c39a594d14bed3f201b24ec0f8.

## Results

The First Settler Effect captured by the American Nations Model demonstrates several significant differences in both traditional and existential wellness among the culturally heterogeneous regions throughout North America. For each of the five wellness measures, a corresponding model was run to assess these differences (See Supplemental). To explicitly and concisely test the hypotheses, we move now to the traditional and existential wellness indexes.

### Hypothesis 1

Model 0 regression analysis for the Traditional Wellness Index reveals significant and nuanced relationships between American Nations and wellness outcomes. The within-subjects mixed-effects model revealed a significant interaction between wellness type and American Nation ($\chi^2_{10} = 77.99$, p < .001), indicating that the balance of traditional versus existential well-being varies markedly by nation. Pairwise contrasts showed that in Yankeedom, traditional well-being

significantly exceeded existential well-being ($\beta \pm SE = 0.845 \pm 0.176$, 95% CI [0.500, 1.190], $t_{121} = 4.80$, $p < .0001$), whereas in the Deep South, existential well-being surpassed traditional by $0.459 \pm 0.176$ (95% CI [−0.803, −0.115], $t_{121} = −2.61$, $p = .010$). Greater Appalachia exhibited the largest reversal, with existential exceeding traditional by $0.999 \pm 0.191$ (95% CI [−1.374, −0.624], $t_{121} = −5.24$, $p < .0001$). The Left Coast also showed higher traditional scores relative to existential ($\beta \pm SE = 0.704 \pm 0.321$, 95% CI [0.077, 1.331], $t_{121} = 2.19$, $p = .030$). In contrast, El Norte, the Far West, the Midlands, New France, New Netherland, the Spanish Caribbean, and Tidewater did not exhibit significant differences (all $p > .05$) (Fig 2).

Notably, the Greater Polynesia (GP) shows a significant positive effect on traditional wellness ($\beta \pm SE = 2.271 \pm 0.911$, CI = 0.474 to 4.068, $p = 0.014$), suggesting higher wellness compared to the reference category, Yankeedom (YK). Conversely, Greater Appalachia (GA) has a significant negative association with traditional wellness ($\beta \pm SE = −0.987 \pm 0.293$, CI = −1.569 to −0.405, $p = 0.001$), indicating lower traditional wellness in this region. The Deep South (DS) also approaches significance with a negative effect ($\beta \pm SE = −0.547 \pm 0.281$, CI = −1.103 to 0.009, $p = 0.055$). Other regions such as El Norte (EN), Far West (FW), and the Left Coast (LC) do not show significant differences from Yankeedom in terms of traditional wellness (Fig 3). Overall, this model highlights the variability in traditional wellness across different American Nations, underscoring the influence of regional cultural and historical contexts. The regression analysis for the Existential Wellness Index highlights several significant predictors among the American Nations and other variables. The Deep South (DS) demonstrates a positive effect on existential wellness ($\beta \pm SE = 0.792 \pm 0.350$, CI = 0.100 to 1.484, $p = 0.026$), indicating higher wellness compared to the reference category, Yankeedom (YK). El Norte (EN) shows a highly significant positive effect ($\beta \pm SE = 1.781 \pm 0.375$, CI = 1.035 to 2.527, $p < 0.001$), suggesting that individuals in this region report higher existential wellness. The Far West (FW) also exhibits a significant positive association ($\beta \pm SE = 0.938 \pm 0.349$, CI = 0.245 to 1.631, $p = 0.009$), as does Greater Appalachia (GA) ($\beta \pm SE = 0.749 \pm 0.327$, CI = 0.099 to 1.399, $p = 0.024$). The Spanish Caribbean (SC) stands out with a notably high positive effect on existential wellness ($\beta \pm SE = 2.096 \pm 0.651$, CI = 0.803 to 3.389, $p = 0.002$) (Fig 3).

In Fig 4, the Model 1 regression analysis examining the Traditional Wellness Index using the American Nations Model reveals several key insights. The coefficient for the Spanish Caribbean (AN: SC) is significant and positive ($\beta \pm SE = 1.305 \pm 0.636$, CI = 0.056 to 2.555, $p = 0.043$), indicating that individuals in this region report higher traditional

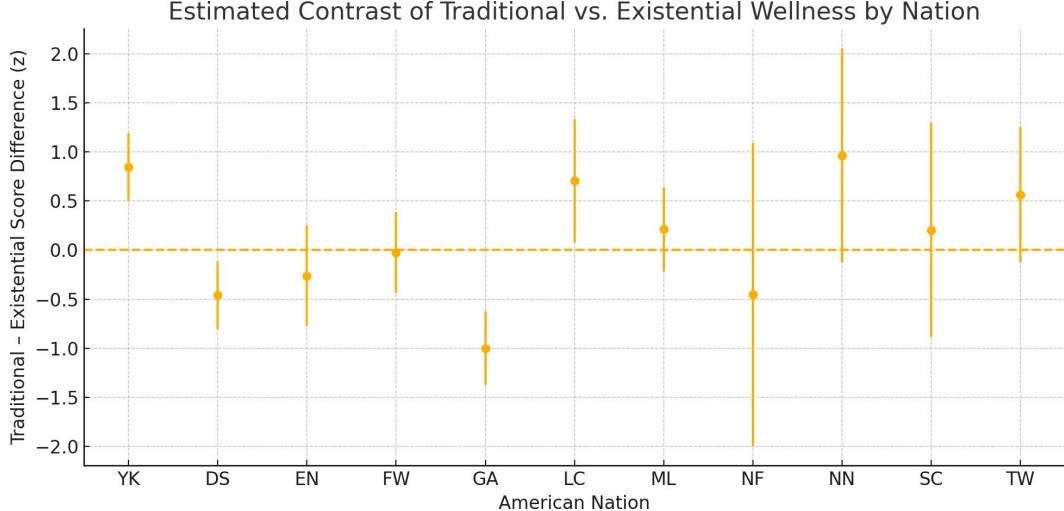

**Fig 2. Within-Nation Mixed-Effects Model.** Nation-specific contrasts with 95% confidence intervals. This pattern indicates that several American Nations—particularly those with honor-based or frontier legacies—display proportionally greater existential versus traditional wellness compared to Yankeedom, highlighting nation-specific tradeoffs.

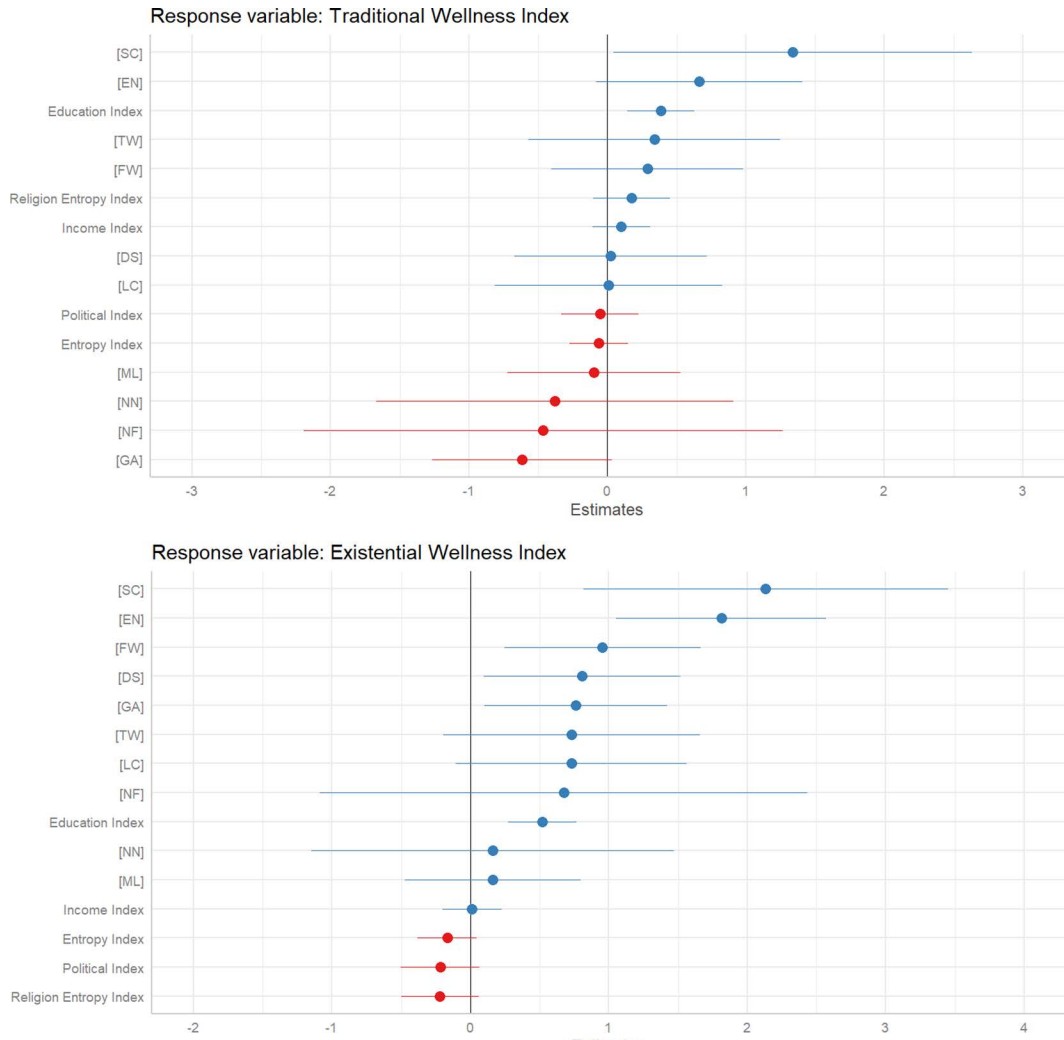

**Fig 3. Fixed Effects Plot for Drivers of Traditional and Existential Wellness: This figure displays fixed effects estimates from a linear model, showing the influence of various factors on traditional wellness (top) and existential wellness (bottom) across the American Nations.** The results indicate that regions like Greater Polynesia and Spanish Caribbean exhibit higher traditional wellness, while regions like the Deep South and El Norte show elevated existential wellness.

wellness compared to the reference category, Yankeedom. Conversely, the coefficient for the Deep South (AN: DS) is negative and approaches significance ($\beta \pm SE = -0.603 \pm 0.319$, $CI = -1.236$ to $0.030$, $p = 0.062$), suggesting lower traditional wellness, though the confidence interval includes zero. The Education Index is also a significant positive predictor ($\beta \pm SE = 0.371 \pm 0.117$, $CI = 0.140$ to $0.602$, $p = 0.002$), indicating that higher education levels are associated with increased traditional wellness. Other predictors, such as the Entropy Index, Political Index, Religion Entropy Index, and Income Index, do not exhibit significant relationships with traditional wellness, as their confidence intervals overlap zero.

In Fig 4, the regression analysis for the Existential Wellness Index using the American Nations Model highlights several significant predictors. The Spanish Caribbean (AN: SC) has a notably positive effect on existential wellness ($\beta \pm SE = 2.096 \pm 0.651$, $CI = 0.803$ to $3.389$, $p = 0.002$), indicating higher wellness compared to the reference category,

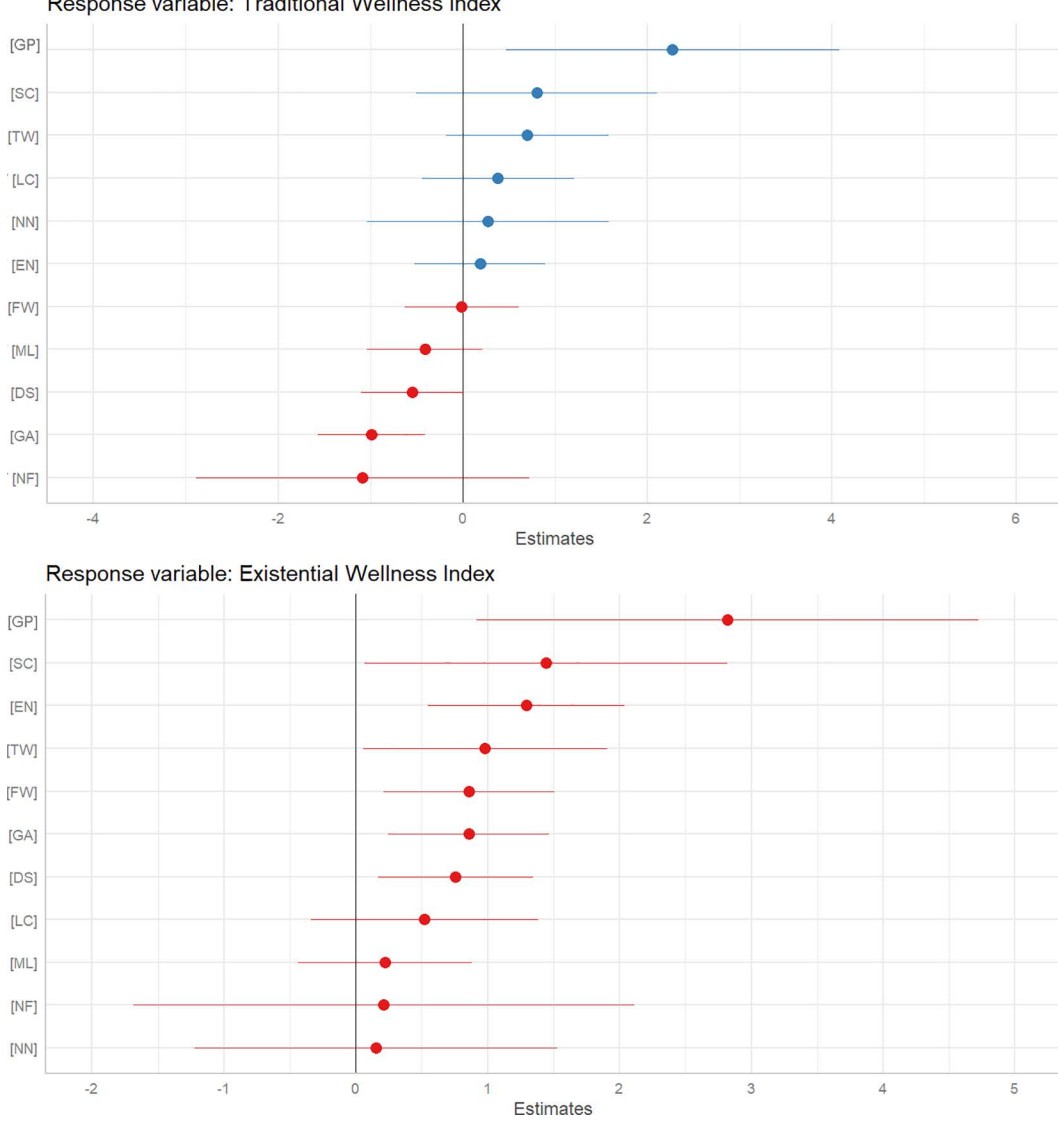

**Fig 4. Fixed Effects Plot for Honor vs. Dignity Cultures on Wellness: This figure displays fixed effects estimates from a linear model, comparing the impact of Honor and Dignity cultures on traditional wellness (top) and existential wellness (bottom) across the American Nations.** The analysis reveals that Honor cultures, particularly in the Southern nations, are associated with higher existential wellness, while Dignity cultures in the Northeastern and Midwestern nations show no significant advantage in either wellness dimension.

Yankeedom. Similarly, El Norte (AN: EN) shows a significant positive effect (β ± SE = 1.781 ± 0.375, CI = 1.035 to 2.527, p < 0.001), suggesting that individuals in this region report higher existential wellness. Other regions, including the Far West (AN: FW), Deep South (AN: DS), and Greater Appalachia (AN: GA), also demonstrate positive and significant associations with existential wellness. However, the coefficients for factors like Entropy Index, Political Index, Religion Entropy Index, and Income Index are not significant, indicating no substantial effect on existential wellness. The Education Index is a significant positive predictor (β ± SE = 0.505 ± 0.120, CI = 0.267 to 0.743, p < 0.001), underscoring the role of education in enhancing existential wellness.

## Hypothesis 2

The regression for the regional analysis of the Traditional Wellness Index identifies significant and non-significant predictors related to different regions (Table 2). The model explains about 36% of the variance in traditional wellness (R-squared = 0.359) and has a significant overall fit (F(8, 100) = 6.993, p < 0.001). The Education Index stands out as a significant positive predictor (β ± SE = 0.403 ± 0.117, CI = 0.171 to 0.634, p = 0.001), indicating higher education levels are associated with increased traditional wellness. However, other predictors, where groups of nations compared against the Northeastern and Midwestern Nations, such as the Southern region (β ± SE = −0.214 ± 0.227, CI = −0.664 to 0.236, p = 0.348), Western region (β ± SE = 0.365 ± 0.228, CI = −0.087 to 0.818, p = 0.112), and Small regions (β ± SE = 1.305 ± 0.620, CI = 0.073 to 2.538, p = 0.038), do not show significant effects on traditional wellness. Additionally, variables such as Entropy Index (β ± SE = −0.007 ± 0.100, CI = −0.205 to 0.191, p = 0.947), Political Index (β ± SE = −0.045 ± 0.118, CI = −0.278 to 0.188, p = 0.704), Religious Entropy Index (β ± SE = 0.121 ± 0.130, CI = −0.137 to 0.379, p = 0.352), and Income Index (β ± SE = 0.059 ± 0.098, CI = −0.135 to 0.253, p = 0.549) also do not show significant associations with traditional wellness.

The regression for regional analysis of the Existential Wellness Index reveals several significant predictors among the regions of nations (Table 3). The model explains about 36% of the variance in existential wellness (R-squared = 0.358) and has a significant overall fit (F(8, 100) = 6.963, p < 0.001). The Southern region Honor Cultures (β ± SE = 0.590 ± 0.229, CI = 0.137 to 1.043, p = 0.011) and the Western region (β ± SE = 1.048 ± 0.230, CI = 0.594 to 1.502, p < 0.001) show positive associations with existential wellness indicating higher existential wellness in these regions compared to the reference category of the Northeastern and Midwestern nations associated with Dignity Cultures. The Small regions also exhibit a significant positive effect (β ± SE = 1.901 ± 0.625, CI = 0.663 to 3.139, p = 0.003). Education is a significant positive predictor (β ± SE = 0.517 ± 0.118, CI = 0.284 to 0.751, p < 0.001), highlighting the importance of education in enhancing existential wellness. Political orientation exhibits a marginally significant negative effect (β ± SE = −0.229 ± 0.119, CI = −0.464 to 0.005, p = 0.057), suggesting that increased political conservatism correlates with lower existential wellness. Variables such as Entropy Index (β ± SE = −0.107 ± 0.101, CI = −0.307 to 0.093, p = 0.289), Religious Entropy Index (β ± SE = −0.254 ± 0.131, CI = −0.513 to 0.005, p = 0.055), and Income Index (β ± SE = −0.058 ± 0.099, CI = −0.254 to 0.138, p = 0.562) do not show significant associations with existential wellness.

**Table 2. Linear regression model summary for the regional comparison of the Traditional Wellness Index. The model includes the predictors: Region (with Northeastern and Midwestern nations as the reference category). The table presents the estimated coefficients, standard errors (S.E.), 95% confidence intervals, and p-values for each predictor.**

| Predictor | Estimate (S.E.) | Confidence Interval | P value |
|---|---|---|---|
| (Intercept) | −0.046 (0.157) | (−0.356, 0.265) | 0.772 |
| Region: Southern | −0.214 (0.227) | (−0.664, 0.236) | 0.348 |
| Region: Western | 0.365 (0.228) | (−0.087, 0.818) | 0.112 |
| Region: Small | 1.305 (0.620) | (0.073, 2.538) | 0.038* |
| Entropy Index | −0.007 (0.100) | (−0.205, 0.191) | 0.947 |
| Political Index | −0.045 (0.118) | (−0.278, 0.188) | 0.704 |
| Education Index | 0.403 (0.117) | (0.171, 0.634) | 0.001** |
| Religion Entropy Index | 0.121 (0.130) | (−0.137, 0.379) | 0.352 |
| Income Index | 0.059 (0.098) | (−0.135, 0.253) | 0.549 |

Significance codes: *** p < 0.001, ** p < 0.01, * p < 0.05.

**Table 3. Linear regression model summary for the regional comparison of the Existential Wellness Index. The model includes the predictors: Region (with Northeastern and Midwestern nations as the reference category). The table presents the estimated coefficients, standard errors (S.E.), 95% confidence intervals, and p-values for each predictor.**

| Predictor | Estimate (S.E.) | Confidence Interval | P value |
|---|---|---|---|
| (Intercept) | −0.565 (0.158) | (−0.876, −0.254) | 0.001** |
| Region: Southern | 0.590 (0.229) | (0.137, 1.043) | 0.011* |
| Region: Western | 1.048 (0.230) | (0.594, 1.502) | <0.001*** |
| Region: Small | 1.901 (0.625) | (0.663, 3.139) | 0.003** |
| Entropy Index | −0.107 (0.101) | (−0.307, 0.093) | 0.289 |
| Political Index | −0.229 (0.119) | (−0.464, 0.005) | 0.057 |
| Education Index | 0.517 (0.118) | (0.284, 0.751) | <0.001*** |
| Religion Entropy Index | −0.254 (0.131) | (−0.513, 0.005) | 0.055 |
| Income Index | −0.058 (0.099) | (−0.254, 0.138) | 0.562 |

Significance codes: *** $p < 0.001$, ** $p < 0.01$, * $p < 0.05$.

## Hypothesis 3

The linear mixed model analysis assessing traditional wellness included race, region, their interaction, and the income index as predictors, with MSA as a random effect (Fig 5). Black residents exhibited significantly lower traditional wellness compared to White residents ($\beta \pm SE = -1.745 \pm 0.399$, $p < 0.001$). However, the interactions between race and region did not reveal substantial differences. The interaction between Black race and the Southern region was positive but not statistically significant ($\beta \pm SE = 0.760 \pm 0.500$, $p = 0.131$), suggesting no significant regional mitigation of racial disparities in traditional wellness. Interestingly, Hispanic residents in the Western region showed a significant positive interaction effect, indicating higher traditional wellness in this group compared to their counterparts in other regions ($\beta \pm SE = 1.154 \pm 0.562$, $p = 0.044$).

The Small nations region exhibited an overall positive effect on traditional wellness ($\beta \pm SE = 1.493 \pm 0.470$, $p = 0.002$), suggesting that residents in these areas, regardless of race, experience higher traditional wellness compared to the reference region. The income index remained a significant positive predictor of traditional wellness ($\beta \pm SE = 0.646 \pm 0.114$, $p < 0.001$), underscoring the critical role of economic stability in overall wellbeing. These results suggest that, after controlling for income, regional cultural factors, whether Honor versus Dignity cultures, do not significantly drive differences in wellness among racial groups within their respective regions, except for the noted positive effect in the Western region for Hispanics and the generally higher wellness in the Small nations. The linear mixed model analysis assessing the Existential Wellness Index offered detailed insights into the Minority Identity Disparity hypothesis. The model included race, region, their interaction, and the income index as predictors, with MSA as a random effect. Black residents exhibited significantly lower existential wellness compared to White residents ($\beta \pm SE = -2.541 \pm 0.381$, $p < 0.001$). Notably, the interaction between Black race and the Southern region showed a significant positive effect ($\beta \pm SE = 1.864 \pm 0.408$, $p < 0.001$), suggesting that Black residents in the Southern nations experience higher existential wellness compared to those in the Northeastern and Midwestern nations. Similarly, the interaction for Hispanic residents in the Southern region was also significant ($\beta \pm SE = 1.630 \pm 0.492$, $p = 0.001$), indicating that Hispanics in the Southern nations report better existential wellness compared to their counterparts in the Northeastern and Midwestern regions. Additionally, Black residents in the Small nations region showed a significant positive interaction, reflecting higher existential wellness in these areas ($\beta \pm SE = 1.447 \pm 0.651$, $p = 0.027$). Interactions for Hispanic and Black residents in the Western region were positive but did not reach statistical significance. Interestingly, the income index was not a significant predictor of existential wellness ($\beta \pm SE = 0.001 \pm 0.125$, $p = 0.998$), highlighting the prominent role of regional and cultural factors over economic stability in shaping existential wellness outcomes.

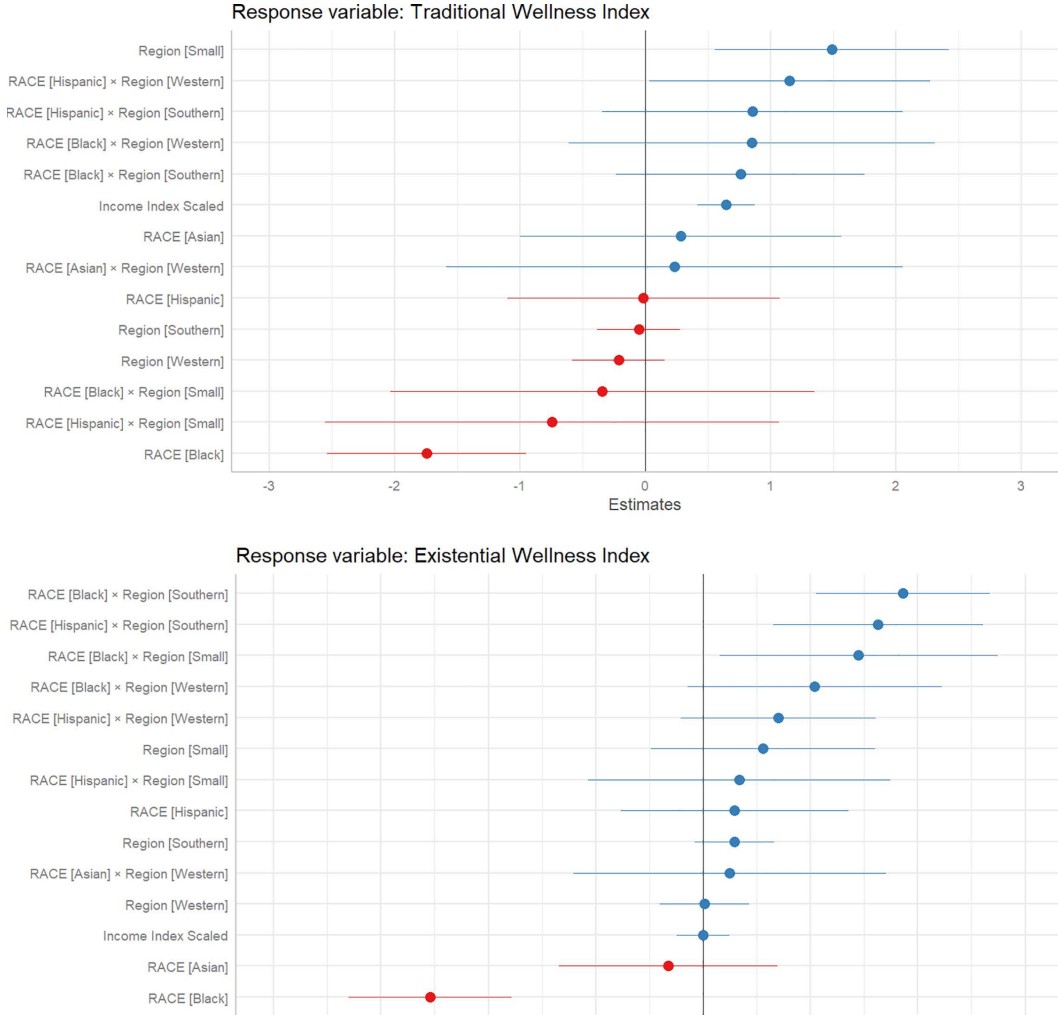

**Fig 5. Fixed Effects Plot for Racial Disparities in Wellness:** This figure illustrates the fixed effects estimates from a linear mixed effects model, highlighting the influence of race on traditional wellness (top) and existential wellness (bottom) across the American Nations. The analysis reveals significant disparities, with Black residents generally reporting lower wellness levels. However, regional interactions indicate that Black and Hispanic residents in Southern and Small Nations experience relatively higher existential wellness, challenging the expected regional disparities.

## Discussion

First, in strong support of the First **Settler Wellness Effect hypothesis**, the American Nations Model demonstrates significant differences in both traditional and existential wellness among the culturally heterogeneous regions throughout the United States. Our analysis reveals that, compared to Yankeedom, Greater Polynesia and Spanish Caribbean exhibits higher traditional wellness, while regions such as Greater Appalachia and the Deep South show lower traditional wellness. In terms of existential wellness, the Deep South, El Norte, the Far West, and Greater Appalachia report higher levels of existential wellness compared to Yankeedom, with the Spanish Caribbean showing notably high existential wellness. These findings underscore the influence of regional, cultural and historical contexts on wellness outcomes, validating the Settler Effect hypothesis and highlighting the variability in wellness across different American Nations. Second, in support

of the **Honor-Based Wellness hypothesis** – the idea that American Nations with honor-based cultural norms will demonstrate greater existential wellbeing, but lower traditional wellbeing – showed that regional cultural factors significantly impact existential wellness, particularly in regions with honor-based cultural norms. Specifically, the Southern Nations of Greater Appalachia, New France, and the Deep South, experience greater existential wellbeing when compared to the Northeastern and Midwestern Nations of Yankeedom, New Netherlands, and the Midlands. Indeed, our within-subjects analysis confirmed that these honor-based regions exhibit a significant reversal in wellness priorities, with existential well-being surpassing traditional well-being—most notably in Greater Appalachia and the Deep South—further reinforcing the cultural divergence in pathways to human flourishing.

Third, we reject the Minority Identity Disparity hypothesis. Our comprehensive analysis of traditional and existential wellness indices reveals that the anticipated regional disparities disadvantaging minority racial groups do not hold. Contrary to expectations, regional differences sometimes had the opposite – a positive effect. While Black residents consistently exhibit lower traditional wellness overall, the interactions between race and region were not significant, suggesting that regional cultural factors do not meaningfully alter these disparities. Economic stability, as measured by the income index, remains a critical determinant of traditional wellness across all racial and regional contexts. However, in terms of existential wellness, our findings indicate a more complex regional dynamic.

Although Black residents generally report lower levels of existential wellness, the significant positive interaction between Black race and the Southern region underscores that certain cultural factors in the South may indeed foster a more supportive environment for existential wellbeing. Moreover, the fact that Black residents in the Northeastern and Midwestern regions fare particularly poorly in existential wellness, yet experience significantly better outcomes in the Small Nations and Southern regions, suggests that these latter regions provide conditions more conducive to existential wellbeing. This highlights the inadequacies in the Midwestern and Northeastern regions in supporting the existential needs of Black residents, who, in contrast, thrive better in regions where cultural and community support structures may be more robust.

Presumably, this relationship could be attributed to stronger social networks and institutions. For example, the African Methodist Episcopal (AME) Church, founded in 1816 by Richard Allen as a protest against racial discrimination and intolerance in white Methodist congregations, likely represents one putative case [53]. Namely, the AME Church provides a supportive community that may alleviate much of this past and current prejudice-related suffering experienced by Black individuals in the Northeast and Midland nations. In particular, the AME Church includes, as part of its unique mission statement, the intention to provide continuing programs to enhance the complete social development of all people within the church, while welcoming and including further church membership among those of diverse ethnicities. Hispanic residents in the Southern region also report higher existential wellness, indicating potential regional cultural benefits. This might be due to many Hispanics being relatively new to the Deep South and potentially self-selected as "pioneer emigrants" pursuing economic opportunities in specific locales. Alternatively, this could also be related to stronger cultural norms around traditional family values, along with their emphasis on intergenerational dwelling, often related to strong adherence to traditional religious practices. However, the absence of significant interaction effects in the Western region suggests that these benefits are not uniformly distributed. Overall, and surprisingly, these results challenge the Minority Identity Disparity hypothesis, demonstrating that while racial disparities in wellness persist, they are not significantly moderated by regional cultural factors along the lines of the ANM.

That said, such factors, undoubtedly, also do not influence wellbeing in isolation, but are likely shaped by various socio-economic and socio-cultural forces, such as education, income, and even feelings of meaning and purpose. For instance, education facilitates socialization by instilling awareness of prevailing cultural and social norms, and shaping appropriate behaviors across diverse contexts. Education, especially early childhood education, further fosters productive engagement in daily activities, including the facilitation of social stability and social bonding [54].

By facilitating engagement and integration into society, education fosters economic growth while diminishing poverty. Education further enables workers to enhance their skills, thereby improving the quality of goods and services produced, ultimately fostering prosperity and enhancing economic competitiveness [59,60]. In particular, public education has long been regarded as a long-term investment in society as a whole, with primary education proving to have particularly high rates of socio-economic return [59,61]. Moreover, apart from bolstering economic prosperity, education contributes to technological and scientific advancements, reduced unemployment, and overall improvements in social equality [62]. Finally, education fosters personal development, including learning new skills, honing talents, nurturing creativity, enhancing self-knowledge, and refining problem-solving and decision-making capabilities [63,64]. Specifically, educated individuals are often more informed about health care issues, benefit from stronger social support networks, and have better coping strategies, all of which allow them to adjust their wellbeing accordingly [65].

Nonetheless, as mentioned previously, income and happiness appear linked and this could be associated with increased leisure time. Leisure time activities that are physical, relational, and performed outdoors are further associated with greater happiness during periods of free time [66]. In fact, quantitative analyses from over thirty different countries show that individuals who prioritize social relationships or who focus on personal development during leisure time activities are generally happier than others [69].

In contrast, it often seems to be internal factors, such as the Big Five Personality Traits, which tend to be more substantially associated with life satisfaction and happiness levels (exempting the Big Five dimension of openness to experience) [70]. Interestingly, the Big Five personality dimensions are associated with the American Nations Model, and heterogeneously distributed among the American nations, and the regional coalitions they represent throughout the United States [71]. For instance, research has shown that certain nations, such as the Deep South and New Netherland, are distinctly characterized by traits linked to authoritarian conventionalism, while others like the Left Coast and Yankeedom score higher on traits related to cognitive resilience and openness. These regional variations underscore how personality traits, aggregated at the community level, contribute to the unique lived experiences across different American nations, revealing a deeper connection between historical settlement patterns and contemporary psychological landscapes.

One intriguing finding from this study was that Honor Cultures of the Southern nations appear to be characterized by greater existential wellness than the Northeastern and Midwestern nations. This phenomenon may be, in part, a result of an evolutionary mismatch in the latter nations, where less emphasis is put on communal identity than individual identity [72]. In other words, within small-scale, indigenous communities, peoples and local communities, happiness is often derived from family relations, social participation, and connections with nature and spiritual beings, as opposed to an often more Western-focused inclination toward greater and more varied material acquisitions [73,74].

Specifically, one recent study examined the relationship between income and global measures of wellness, challenging the prevailing assumption that high life satisfaction is exclusive to wealthy societies [5]. The research surveyed 2,966 members of Indigenous Peoples and local communities across 19 globally distributed sites. The findings reveal that many populations with very low monetary incomes report high average levels of life satisfaction, comparable to those in high-income countries. This study challenges the prevailing belief that material wealth is essential for high levels of wellness, and that non-material factors are also often equally as important for enhancing wellness – such as a strong sense of community identity, as well as a strong feeling of purpose and meaning. Specifically, social support, trust, and freedom are critical elements, along with the absence of corruption. This important study may provide insights directly relevant to our findings, where a nation like Greater Appalachia – characterized by less trust of institutions and authorities, greater trust in local communities, and more direct interaction with land-based subsistence (when directly compared to Yankeedom) – may have resultant lower traditional wellness but greater existential wellness [75].

Moreover, the same trend appears to apply not only to small-scale traditional societies, but still further across much larger nation-states. In particular, Finland has been consistently ranked as the happiest country in the world for seven

years in a row according to a joint academic partnership between Gallup, the Wellbeing Research Centre at the University of Oxford, and the UN Sustainable Development Solutions Network's World Happiness Report [76]. According to the Organization for Economic Cooperation and Development (OECD) Better Life Index, the fact that Finland outperforms the average country in happiness levels, appears to be attributable primarily not to the acquisition of material resources, but rather to combined socio-ecological factors, such as education, a substantive social safety net, social trust, freedom, gender equality, a sense of community, public services, and access to nature [77].

The American Nations framework provides a valuable tool for understanding regional differences within the United States and their influence on a wide range of sociocultural phenomena. That said, regional (or "national") cultural differences are just one factor among many influences at work in human development and individual behavior, including interactions between different groups, environmental factors, and evolving social and political dynamics.

Future research should build on these findings by empirically testing the psychosocial mechanisms that may underlie the observed relationships between regional culture and wellness. In particular, mediation models examining whether variables such as institutional trust and social connection serve as pathways between regional identity and wellness outcomes would offer critical insight into how historical cultural legacies continue to shape present-day wellbeing. While the current study is limited by its use of aggregate, MSA-level data—precluding formal mediation analysis—future studies using individual-level survey data or longitudinal panel designs could rigorously test these pathways. Such work would allow researchers to model temporal sequencing and test for causal mechanisms, ultimately enriching our understanding of how cultural and institutional factors mediate the link between regional context and dimensions of human flourishing.

Ultimately, what emerging research now suggests, is that what appears to have the most substantive influence on happiness and wellbeing are often internal factors such as genetics, personality traits, and having a high internal locus of control [78–80]. That said, with such apparently strong internal influences on happiness, it is truthfully often quite difficult to have a significant external influence on happiness and wellbeing. In this sense, establishing and especially promoting any sort of happiness metric, whether scientifically validated or not, has often been argued to be a substantively useful exercise and opportunity for political gain. In fact, some scholars have argued that national aggregate levels of subjective wellbeing can often account for even more variance in government vote share than standard macroeconomic variables, such as income and rates of employment [81].

## Conclusion

In conclusion, our study underscores the profound influence that cultural and regional factors exert on both traditional and existential wellness across the U.S. regional cultures identified in the American Nations model. While traditional measures of wellness, such as economic stability and health, remain crucial, our findings highlight the importance of existential dimensions, such as community identity and a sense of purpose, particularly within regions with honor-based cultural norms. The variability in wellness outcomes across different American Nations challenges the conventional belief that material wealth alone is the primary driver of well-being. Instead, our results suggest that non-material factors—such as social support, trust, freedom, and a strong sense of community—play an equally significant, if not greater, role in fostering high levels of life satisfaction and wellness.

This holistic approach to understanding wellness aligns with the broader goal of advancing human welfare as outlined in the World Health Organization's Constitution [81], which recognizes health as encompassing complete physical, mental, and social well-being. As scientific understanding of wellness continues to evolve, it becomes increasingly clear that promoting well-being in its fullest sense—beyond just the absence of disease—requires a comprehensive strategy that integrates cultural, psychological, and social factors. By embracing this broader definition of wellness, societies can move closer to realizing the vision of universal well-being, where all individuals, regardless of their cultural or regional background, can achieve a state of flourishing and fulfillment.

## Supporting information

**S1 Fig. Physical wellbeing.** Summary plot of physical wellbeing scores across MSAs in the U.S.
(TIF)

**S2 Fig. Social wellbeing.** Summary plot of social wellbeing scores across MSAs in the U.S.
(TIF)

**S3 Fig. Financial wellbeing.** Summary plot of financial wellbeing scores across MSAs in the U.S.
(TIF)

**S4 Fig. Community wellbeing.** Summary plot of community wellbeing scores across MSAs in the U.S.
(TIF)

**S5 Fig. Purpose wellbeing.** Summary plot of purpose wellbeing scores across MSAs in the U.S.
(TIF)

## Acknowledgments

We thank the two reviewers and the editor for comments that substantially improved the quality of the manuscript. We also thank the Department of Anthropology, University of Toronto Mississauga for further support.

## Author contributions

**Conceptualization:** David R. Samson, Colin Woodard.

**Data curation:** David R. Samson.

**Formal analysis:** David R. Samson.

**Funding acquisition:** David R. Samson.

**Investigation:** Colin Woodard.

**Methodology:** David R. Samson, Nathan Oesch, Colin Woodard.

**Project administration:** Nathan Oesch.

**Resources:** Nathan Oesch.

**Writing – original draft:** David R. Samson, Nathan Oesch, Colin Woodard.

**Writing – review & editing:** Colin Woodard.

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
