## [Decision Letter · Decision Letter 0]

9 Apr 2025

PONE-D-24-60508Wellbeing Across the American Nations: First Settler Effects influence traditional and existential wellnessPLOS ONE

Dear Dr. Samson,

Thank you for submitting your manuscript to PLOS ONE. After careful consideration, we feel that it has merit but does not fully meet PLOS ONE’s publication criteria as it currently stands. Therefore, we invite you to submit a revised version of the manuscript that addresses the points raised during the review process.

**Both Reviewers appreciated the manuscript and encouraged its publication, albeit suggesting relevant and substantial changes. Even Reviewer #1, who submitted a judgment of acceptance of the manuscript, then in fact in the commentary moved an important methodological issue, which requires at least a clarifying response from the authors. Reviewer #2 offers a complex body of comments, requiring a thorough review of the study.**

We look forward to receiving your revised manuscript.

Kind regards,

Stefano Federici, Ph.D.

Academic Editor

PLOS ONE

**Journal Requirements:**

1. When submitting your revision, we need you to address these additional requirements. Please ensure that your manuscript meets PLOS ONE's style requirements, including those for file naming. The PLOS ONE style templates can be found at https://journals.plos.org/plosone/s/file?id=wjVg/PLOSOne_formatting_sample_main_body.pdf and https://journals.plos.org/plosone/s/file?id=ba62/PLOSOne_formatting_sample_title_authors_affiliations.pdf 2. Thank you for stating in your Funding Statement: This research was supported through Discovery Grants from the Natural Sciences and Engineering Research Council of Canada (RGPIN-2020-05942 to D.R.S.). Additional support was provided by the Department ofAnthropology, University of Toronto Mississauga and The Pell Center’s Nationhood Lab.  Please provide an amended statement that declares *all* the funding or sources of support (whether external or internal to your organization) received during this study, as detailed online in our guide for authors at http://journals.plos.org/plosone/s/submit-now.  Please also include the statement “There was no additional external funding received for this study.” in your updated Funding Statement. Please include your amended Funding Statement within your cover letter. We will change the online submission form on your behalf. 3. Thank you for stating the following in the Acknowledgments Section of your manuscript: We thank the funding agency that has helped support this research through Discovery Grants from the Natural Sciences and Engineering Research Council of Canada (RGPIN-2020-05942 to D.R.S). We also thank the Department of Anthropology, University of Toronto Mississauga for further support. We note that you have provided funding information that is not currently declared in your Funding Statement. However, funding information should not appear in the Acknowledgments section or other areas of your manuscript. We will only publish funding information present in the Funding Statement section of the online submission form. Please remove any funding-related text from the manuscript and let us know how you would like to update your Funding Statement. Currently, your Funding Statement reads as follows: This research was supported through Discovery Grants from the Natural Sciences and Engineering Research Council of Canada (RGPIN-2020-05942 to D.R.S.). Additional support was provided by the Department of Anthropology, University of Toronto Mississauga and The Pell Center’s Nationhood Lab Please include your amended statements within your cover letter; we will change the online submission form on your behalf. 4. When completing the data availability statement of the submission form, you indicated that you will make your data available on acceptance. We strongly recommend all authors decide on a data sharing plan before acceptance, as the process can be lengthy and hold up publication timelines. Please note that, though access restrictions are acceptable now, your entire data will need to be made freely accessible if your manuscript is accepted for publication. This policy applies to all data except where public deposition would breach compliance with the protocol approved by your research ethics board. If you are unable to adhere to our open data policy, please kindly revise your statement to explain your reasoning and we will seek the editor's input on an exemption. Please be assured that, once you have provided your new statement, the assessment of your exemption will not hold up the peer review process. 5. Your ethics statement should only appear in the Methods section of your manuscript. If your ethics statement is written in any section besides the Methods, please move it to the Methods section and delete it from any other section. Please ensure that your ethics statement is included in your manuscript, as the ethics statement entered into the online submission form will not be published alongside your manuscript. 6. We note that Figure 1 in your submission contain map images which may be copyrighted. All PLOS content is published under the Creative Commons Attribution License (CC BY 4.0), which means that the manuscript, images, and Supporting Information files will be freely available online, and any third party is permitted to access, download, copy, distribute, and use these materials in any way, even commercially, with proper attribution. For these reasons, we cannot publish previously copyrighted maps or satellite images created using proprietary data, such as Google software (Google Maps, Street View, and Earth). For more information, see our copyright guidelines: http://journals.plos.org/plosone/s/licenses-and-copyright. We require you to either present written permission from the copyright holder to publish these figures specifically under the CC BY 4.0 license, or remove the figures from your submission: a. You may seek permission from the original copyright holder of Figure 1 to publish the content specifically under the CC BY 4.0 license.   We recommend that you contact the original copyright holder with the Content Permission Form (http://journals.plos.org/plosone/s/file?id=7c09/content-permission-form.pdf) and the following text:“I request permission for the open-access journal PLOS ONE to publish XXX under the Creative Commons Attribution License (CCAL) CC BY 4.0 (http://creativecommons.org/licenses/by/4.0/). Please be aware that this license allows unrestricted use and distribution, even commercially, by third parties. Please reply and provide explicit written permission to publish XXX under a CC BY license and complete the attached form.” Please upload the completed Content Permission Form or other proof of granted permissions as an "Other" file with your submission. In the figure caption of the copyrighted figure, please include the following text: “Reprinted from [ref] under a CC BY license, with permission from [name of publisher], original copyright [original copyright year].” b. If you are unable to obtain permission from the original copyright holder to publish these figures under the CC BY 4.0 license or if the copyright holder’s requirements are incompatible with the CC BY 4.0 license, please either i) remove the figure or ii) supply a replacement figure that complies with the CC BY 4.0 license. Please check copyright information on all replacement figures and update the figure caption with source information. If applicable, please specify in the figure caption text when a figure is similar but not identical to the original image and is therefore for illustrative purposes only.The following resources for replacing copyrighted map figures may be helpful: USGS National Map Viewer (public domain): http://viewer.nationalmap.gov/viewer/The Gateway to Astronaut Photography of Earth (public domain): http://eol.jsc.nasa.gov/sseop/clickmap/Maps at the CIA (public domain): https://www.cia.gov/library/publications/the-world-factbook/index.html and https://www.cia.gov/library/publications/cia-maps-publications/index.htmlNASA Earth Observatory (public domain): http://earthobservatory.nasa.gov/Landsat:
http://landsat.visibleearth.nasa.gov/USGS EROS (Earth Resources Observatory and Science (EROS) Center) (public domain): http://eros.usgs.gov/#Natural Earth (public domain): http://www.naturalearthdata.com/

**Additional Editor Comments:**

Both Reviewers appreciated the manuscript and encouraged its publication, albeit suggesting relevant and substantial changes. Even Reviewer #1, who submitted a judgment of acceptance of the manuscript, then in fact in the commentary moved an important methodological issue, which requires at least a clarifying response from the authors. Reviewer #2 offers a complex body of comments, requiring a thorough review of the study.

Reviewers' comments:

Reviewer's Responses to Questions

**Comments to the Author**

1. Is the manuscript technically sound, and do the data support the conclusions?

Reviewer #1: Yes

Reviewer #2: Yes

2. Has the statistical analysis been performed appropriately and rigorously? 

Reviewer #1: I Don't Know

Reviewer #2: Yes

3. Have the authors made all data underlying the findings in their manuscript fully available?

Reviewer #1: Yes

Reviewer #2: Yes

4. Is the manuscript presented in an intelligible fashion and written in standard English?

Reviewer #1: Yes

Reviewer #2: Yes

5. Review Comments to the Author

**Reviewer #1:**  The submission is a very interesting piece that does contribute to the important topic of historical-and-cultural prerequisites for regional devision of the United States. The hypotheses the authors test and two of which verify are really important for a better understanding of the differences between the US historical and cultural macroregions - the North and the South (divided into smaller regions by the paper authors). The research shows convincingly (although I am not a specilist in statistical methods) that this division finds a remarkable manifestation in significant, even fundamental differences in the Northerners and Southerners' wellbeing. So, the paper is good and definitely deserves publication. Yet my final comment would be that it could become even better if the authors had done at least several interviews with people from different American nations. People's explanations of their views and state of mind, "voices from the field", could make the paper less "dry" in its style of presentation, this could allow the reader to see just human beings behind the figures.

**Reviewer #2:**  Dear Authors,

Thank you for writing such a fascinating paper. Here, I believe this paper should be published, though it has a few major and minor issues that I believe should be addressed to substantially improve the manuscript.

Major Issues:

The manuscript repeatedly references the $75,000 income plateau in well-being (Kahneman & Deaton, 2010). However, more recent research by Killingsworth (2021), which used real-time, continuous measures of experienced well-being in a large U.S. sample, found a linear relationship with income across the full range, including well above $75,000. A revised introduction/discussion should acknowledge this recent research and discuss the nuanced relationship between material wealth and wellbeing accordingly.

The manuscript introduces the First Settler Wellness Effect and asserts that early settlement patterns explain current wellness differences, but it does not sufficiently elaborate the psychological mechanisms underlying this persistence. Specifically, the psychological or institutional processes through which regional cultures continue to influence wellness outcomes today, despite changes in ecology, economy, and demography, remain vague. For example, while honor cultures are attributed to herding societies, there is no explanation of why those patterns persist now that herding has largely declined. Integrating theories of cultural transmission would help clarify and tighten the hypotheses for how the historical legacies hypothesized translate into contemporary individual-level outcomes.

The classification of social well-being within the Traditional Wellness Index, rather than the Existential one, needs further justification. Social relationships seem quite related to existential wellness. Did the authors empirically test the factor structure of the wellness components to examine their validity? Does social well-being fit better in the traditional wellness index? Demonstrating this would be useful.

The manuscript implies plausible psychological and social mechanisms through which regional culture influences different dimensions of wellness, but these are not formally tested. In particular, two mediation models seem especially warranted: (1) Region → Social Connection → Existential Wellness, which reflects how communal norms and belonging foster purpose and meaning; and (2) Region → Institutional Trust/Strength → Traditional Wellness, which reflects how confidence in institutions (e.g., health care, governance, economic systems) supports material and physical stability. Testing these pathways empirically, through mediation models, would clarify the mechanisms of the First Settler Wellness Effect and strengthen the claims.

Given the conceptual distinction between traditional and existential wellness, it would be informative to test whether the predictors have significantly different effects across these two wellness types. This could be accomplished via a within-subjects mixed-effects model that includes a wellness-type factor and its interactions with the primary predictors. Such an analysis could more directly assess claims regarding the differences or tradeoffs between material stability and existential fulfillment across regions. That is, do the relationships between regions and each wellness type significantly differ from one another?

Minor issues:

The manuscript presents the First Settler Effects hypothesis and the American Nations framework in a way that feels overly deterministic. While the model offers a useful lens, it has also been subject to scholarly critique for its generalizations. A brief acknowledgment of these critiques in the introduction would help temper the stronger claims and reflect the diversity within regions in the U.S.

The manuscript states that “additional research has found a significant association between having a strong internal locus of control and overall happiness,” but no citation is provided for this claim.

While the study’s central findings are compelling and well-supported, the final discussion section would benefit from substantial tightening. Much of the latter half reads as exploratory rather than integrative, with several tangents (e.g., early childhood education, DRAMMA framework, internet use, indigenous communities) that, although interesting, dilute its main focus.

6. PLOS authors have the option to publish the peer review history of their article (what does this mean? ). If published, this will include your full peer review and any attached files.

**Do you want your identity to be public for this peer review?** For information about this choice, including consent withdrawal, please see our Privacy Policy .

Reviewer #1: **Yes: ** Dmitri M. Bondarenko

Reviewer #2: No

---

## [Author Response · Author response to Decision Letter 1]

17 Jun 2025

Response to Reviewers

Reviewer #1:

The submission is a very interesting piece that does contribute to the important topic of historical-and-cultural prerequisites for regional devision of the United States. The hypotheses the authors test and two of which verify are really important for a better understanding of the differences between the US historical and cultural macroregions - the North and the South (divided into smaller regions by the paper authors). The research shows convincingly (although I am not a specilist in statistical methods) that this division finds a remarkable manifestation in significant, even fundamental differences in the Northerners and Southerners' wellbeing. So, the paper is good and definitely deserves publication. Yet my final comment would be that it could become even better if the authors had done at least several interviews with people from different American nations. People's explanations of their views and state of mind, "voices from the field", could make the paper less "dry" in its style of presentation, this could allow the reader to see just human beings behind the figures.

Thank you very much for your generous and thoughtful review. We deeply appreciate your engagement with the manuscript and your recognition of the importance of examining the cultural and historical foundations of regional differences in well-being across the United States. Your comment regarding the value of including “voices from the field” is both insightful and well-taken. We agree that integrating personal narratives could enrich the presentation by humanizing the quantitative results and offering deeper insight into the lived experiences behind the patterns we observe.

However, our primary data source—the Gallup-Healthways Well-Being Index—is based on anonymized responses collected under strict confidentiality protocols. As such, it is not possible for us to identify or contact individual respondents to collect narrative data or conduct follow-up interviews. While this protects participant privacy and facilitates large-scale participation, it limits our ability to directly incorporate testimonial perspectives from individuals within the different American Nations.

That said, we fully agree that future research building on this work would benefit greatly from a mixed-methods approach that includes qualitative interviews alongside population-level survey data. We have noted this important opportunity for future work in the revised discussion.

Once again, we thank you for your constructive feedback and your support of the manuscript.

Reviewer #2:

Dear Authors,

Thank you for writing such a fascinating paper. Here, I believe this paper should be published, though it has a few major and minor issues that I believe should be addressed to substantially improve the manuscript.

Major Issues:

1. The manuscript repeatedly references the $75,000 income plateau in well-being (Kahneman & Deaton, 2010). However, more recent research by Killingsworth (2021), which used real-time, continuous measures of experienced well-being in a large U.S. sample, found a linear relationship with income across the full range, including well above $75,000. A revised introduction/discussion should acknowledge this recent research and discuss the nuanced relationship between material wealth and wellbeing accordingly.

Thank you for this excellent suggestion. In response, we have revised the manuscript to incorporate findings from Killingsworth (2021), which challenge the widely cited $75,000 income plateau by showing that both experienced and evaluative well-being continue to rise with income. This update offers a more accurate and current view of the relationship between income and well-being, and we appreciate your recommendation to include it.

2. The manuscript introduces the First Settler Wellness Effect and asserts that early settlement patterns explain current wellness differences, but it does not sufficiently elaborate the psychological mechanisms underlying this persistence. Specifically, the psychological or institutional processes through which regional cultures continue to influence wellness outcomes today, despite changes in ecology, economy, and demography, remain vague. For example, while honor cultures are attributed to herding societies, there is no explanation of why those patterns persist now that herding has largely declined. Integrating theories of cultural transmission would help clarify and tighten the hypotheses for how the historical legacies hypothesized translate into contemporary individual-level outcomes.

Thank you for this thoughtful and constructive comment. In response, we have revised the manuscript to more clearly articulate the psychological and institutional mechanisms that may account for the persistence of regional cultural influences on wellness. Specifically, we now draw on theories of cultural transmission—including vertical, horizontal, and oblique pathways—to explain how values associated with early settlement patterns (e.g., honor norms rooted in herding economies) can persist across generations even in the absence of their original ecological conditions. We also highlight how institutions such as education systems, religious organizations, and local governance structures act as carriers of these cultural norms, embedding them into daily life and shaping individual-level outcomes over time. These additions help clarify the hypothesized pathways linking historical legacies to contemporary wellness and strengthen the theoretical foundation of the First Settler Wellness Effect.

3. The classification of social well-being within the Traditional Wellness Index, rather than the Existential one, needs further justification. Social relationships seem quite related to existential wellness. Did the authors empirically test the factor structure of the wellness components to examine their validity? Does social well-being fit better in the traditional wellness index? Demonstrating this would be useful.

Thank you for this thoughtful comment. We agree that social well-being could plausibly relate to either traditional or existential wellness, and we carefully considered its classification. However, we chose to group social well-being with traditional wellness based on both theoretical and empirical grounds. Theoretically, the Traditional Wellness Index is designed to capture domains tied to material stability and institutional trust—dimensions that include financial security, physical health, and the strength of everyday social ties that are often maintained through routine, structured interactions (e.g., family, neighbors, workplace networks). In contrast, the Existential Wellness Index emphasizes purpose, meaning, and communal belonging on a more reflective or ideological level.

Thank you for raising this important point. We agree that social well-being occupies a conceptual space that could plausibly align with either traditional or existential wellness. However, we chose to include it in the Traditional Wellness Index based on substantial supporting evidence linking social connectedness to concrete physical and physiological outcomes. Specifically, loneliness and social isolation have been repeatedly associated with increased risk of cardiovascular disease, weakened immune function, cognitive decline, and higher all-cause mortality—patterns more consistent with material and health-related domains than with psychological meaning-making. While social bonds certainly contribute to existential fulfillment, their measurable effects on physical health and survival provide a strong rationale for grouping social well-being alongside other traditional wellness dimensions. We have clarified this justification in the manuscript.:

“Although social well-being could reasonably be associated with existential dimensions, we classified it within the Traditional Wellness Index due to extensive evidence linking the quality of social relationships—particularly loneliness and social isolation—to concrete physical health outcomes such as cardiovascular disease, immune function, and mortality risk, aligning it more closely with other material and physiological dimensions of wellness.”

4. The manuscript implies plausible psychological and social mechanisms through which regional culture influences different dimensions of wellness, but these are not formally tested. In particular, two mediation models seem especially warranted: (1) Region → Social Connection → Existential Wellness, which reflects how communal norms and belonging foster purpose and meaning; and (2) Region → Institutional Trust/Strength → Traditional Wellness, which reflects how confidence in institutions (e.g., health care, governance, economic systems) supports material and physical stability. Testing these pathways empirically, through mediation models, would clarify the mechanisms of the First Settler Wellness Effect and strengthen the claims.

We appreciate the reviewer’s insightful recommendation to test mediation models assessing how regional culture influences wellness through proposed pathways such as social connection and institutional trust. Conceptually, these models align well with the First Settler Wellness Effect framework. They reflect plausible psychosocial mechanisms by which deep-seated cultural norms may shape contemporary well-being—namely, that regional cultural histories influence institutional trust and communal belonging, which in turn affect traditional and existential wellness outcomes. These mediational pathways would significantly enrich our theoretical understanding of how historical legacies exert influence on present-day individual experiences.

However, at a technical level, our current dataset is composed of aggregated wellness scores at the Metropolitan Statistical Area (MSA) level. Mediation models are most appropriately conducted using individual-level data that can account for the variability of both mediators and outcomes within subjects. Aggregated data constrain our ability to make causal inferences about the mediating processes, particularly because assumptions of temporal order, independence, and within-group variability—necessary for mediation—cannot be assured. Additionally, key mediators such as institutional trust are not directly measured in the current dataset.

Given these constraints, we believe implementing these mediation models with the current data would risk violating core statistical assumptions. Nonetheless, we strongly agree with the theoretical importance of testing these pathways. We therefore identify this as a promising avenue for future research using either individual-level survey data or longitudinal panel designs where both mediators and outcomes can be measured directly and temporally ordered. Such work would advance understanding of the psychosocial mechanisms underpinning cultural determinants of wellness. To this end, we add this text to the Discussion:

“Future research should build on these findings by empirically testing the psychosocial mechanisms that may underlie the observed relationships between regional culture and wellness. In particular, mediation models examining whether variables such as institutional trust and social connection serve as pathways between regional identity and wellness outcomes would offer critical insight into how historical cultural legacies continue to shape present-day wellbeing. While the current study is limited by its use of aggregate, MSA-level data—precluding formal mediation analysis—future studies using individual-level survey data or longitudinal panel designs could rigorously test these pathways. Such work would allow researchers to model temporal sequencing and test for causal mechanisms, ultimately enriching our understanding of how cultural and institutional factors mediate the link between regional context and dimensions of human flourishing.”

5. Given the conceptual distinction between traditional and existential wellness, it would be informative to test whether the predictors have significantly different effects across these two wellness types. This could be accomplished via a within-subjects mixed-effects model that includes a wellness-type factor and its interactions with the primary predictors. Such an analysis could more directly assess claims regarding the differences or tradeoffs between material stability and existential fulfillment across regions. That is, do the relationships between regions and each wellness type significantly differ from one another?

In response to the reviewer’s insightful suggestion, we conducted a within-subjects mixed-effects analysis to formally test whether the effects of American Nation on well-being differ between traditional and existential domains. By including wellness type as a factor and modeling its interaction with cultural region, this analysis directly assessed whether the cultural predictors exhibit distinct influences across wellness dimensions. The significant wellness type × nation interaction confirmed that several American Nations—particularly those with honor-based cultural legacies such as Greater Appalachia and the Deep South—demonstrate a clear reversal in wellness profiles, with existential well-being significantly exceeding traditional well-being. These results substantiate the core claim of the Honor-Based Wellness hypothesis: that cultural regions vary not only in overall wellness levels, but in the very nature of what constitutes well-being across contexts. We’ve added language describing the analysis and a Figure (now Fig. 2).

Minor issues:

The manuscript presents the First Settler Effects hypothesis and the American Nations framework in a way that feels overly deterministic. While the model offers a useful lens, it has also been subject to scholarly critique for its generalizations. A brief acknowledgment of these critiques in the introduction would help temper the stronger claims and reflect the diversity within regions in the U.S.

Thank you for this thoughtful comment. We agree that while the American Nations framework offers a valuable lens for interpreting regional cultural differences, it is important to acknowledge its limitations and avoid presenting it as overly deterministic. In response, we have added language to the introduction that explicitly recognizes the dynamic nature of cultural identities and the influence of additional factors such as environmental conditions, institutional change, and individual agency. This addition is intended to temper stronger claims and reflect the diversity and complexity within and across U.S. regions.

The manuscript states that “additional research has found a significant association between having a strong internal locus of control and overall happiness,” but no citation is provided for this claim.

Great catch! It’s been edited.

While the study’s central findings are compelling and well-supported, the final discussion section would benefit from substantial tightening. Much of the latter half reads as exploratory rather than integrative, with several tangents (e.g., early childhood education, DRAMMA framework, internet use, indigenous communities) that, although interesting, dilute its main focus.

Thank you for this helpful observation. In response, we carefully reviewed the entire discussion section and made substantial revisions to improve clarity and focus. Specifically, we streamlined or removed several of the more exploratory tangents, including content on early childhood education, internet use, and peripheral frameworks, in order to keep the emphasis on the central findings and theoretical contributions of the study. These changes, which can be seen in the track changes, reflect our effort to present a more integrative and cohesive conclusion.

---

## [Decision Letter · Decision Letter 1]

25 Jun 2025

Wellbeing Across the American Nations: First Settler Effects influence traditional and existential wellness

PONE-D-24-60508R1

Dear Dr. Samson,

We’re pleased to inform you that your manuscript has been judged scientifically suitable for publication and will be formally accepted for publication once it meets all outstanding technical requirements.

Kind regards,

Stefano Federici, Ph.D.

Academic Editor

PLOS ONE

Additional Editor Comments (optional):

Reviewers' comments:

Reviewer's Responses to Questions

**Comments to the Author**

1. If the authors have adequately addressed your comments raised in a previous round of review and you feel that this manuscript is now acceptable for publication, you may indicate that here to bypass the “Comments to the Author” section, enter your conflict of interest statement in the “Confidential to Editor” section, and submit your "Accept" recommendation.

Reviewer #1: (No Response)

2. Is the manuscript technically sound, and do the data support the conclusions?

Reviewer #1: (No Response)

3. Has the statistical analysis been performed appropriately and rigorously? 

Reviewer #1: (No Response)

4. Have the authors made all data underlying the findings in their manuscript fully available?

Reviewer #1: (No Response)

5. Is the manuscript presented in an intelligible fashion and written in standard English?

Reviewer #1: (No Response)

6. Review Comments to the Author

Reviewer #1: Dear Authors,

Thank you for reacting to my comments. I do understand that your main source of information presupposes anonymity of the collected and processed data. I hope you will employ a mixture of methods which could include your own interviews with people beyond the anonymized database while working on your next paper.

7. PLOS authors have the option to publish the peer review history of their article (what does this mean? ). If published, this will include your full peer review and any attached files.

**Do you want your identity to be public for this peer review?** For information about this choice, including consent withdrawal, please see our Privacy Policy .

Reviewer #1: **Yes: ** Dmitri M. Bondarenko
